# Impact of Exercise Training at Maximal Fat Oxidation Intensity on Metabolic and Epigenetic Parameters in Patients with Overweight and Obesity: Study Protocol of a Randomized Controlled Trial

**DOI:** 10.3390/jfmk9040214

**Published:** 2024-10-31

**Authors:** Marco Antonio Hernández-Lepe, David Alfredo Hernández-Ontiveros, Isaac Armando Chávez-Guevara, Arnulfo Ramos-Jiménez, Rosa Patricia Hernández-Torres, Reymond Josué López-Fregoso, Omar Ramos-Lopez, Francisco José Amaro-Gahete, Raquel Muñiz-Salazar, Francisco Javier Olivas-Aguirre

**Affiliations:** 1Conahcyt National Laboratory of Body Composition and Energetic Metabolism (LaNCoCoME), Tijuana 22390, Mexico; marco.antonio.hernandez.lepe@uabc.edu.mx (M.A.H.-L.); david.hernandez@uabc.edu.mx (D.A.H.-O.); isaac.chavez.guevara@uabc.edu.mx (I.A.C.-G.); aramos@uacj.mx (A.R.-J.); rhernant@uach.mx (R.P.H.-T.); reymond.lopez@uabc.edu.mx (R.J.L.-F.); oscar.omar.ramos.lopez@uabc.edu.mx (O.R.-L.); ramusal@uabc.edu.mx (R.M.-S.); 2Medical and Psychology School, Autonomous University of Baja California, Tijuana 22390, Mexico; 3Faculty of Sports, Campus Ensenada, Autonomous University of Baja California, Ensenada 22800, Mexico; 4Department of Health Sciences, Biomedical Sciences Institute, Autonomous University of Ciudad Juarez, Ciudad Juarez 32310, Mexico; 5Faculty of Physical Culture Sciences, Autonomous University of Chihuahua, Chihuahua 31000, Mexico; 6Department of Physiology, Faculty of Medicine, University of Granada, 18001 Granada, Spain; amarof@ugr.es; 7CIBER of Obesity and Nutrition Pathophysiology (CIBEROBN), National Institute of Health Carlos III, 28029 Granada, Spain; 8Biosanitary Research Institute, Ibs.Granada, 18012 Granada, Spain; 9School of Health Sciences, Autonomous University of Baja California, Ensenada 22860, Mexico

**Keywords:** obesity, overweight, exercise, FatMax

## Abstract

**Background:** Exercise is an essential pillar for human health, as it contributes to physical, mental, and emotional well-being. Well-recognized international organizations, such as the World Health Organization, advocate for integrating exercise into healthy lifestyles, recognizing its importance in disease prevention and improving quality of life. However, despite the consensus on its value, there is no universal agreement on specific prescriptions for vulnerable groups, highlighting the need for personalized approaches that consider the unique characteristics and needs of everyone. Emerging studies have demonstrated that exercise training performed at the intensity that elicits maximal fat oxidation improves insulin sensitivity, cardiorespiratory fitness, and body composition in patients with obesity, making it a highly effective strategy for long-term weight management and metabolic health in this specific population. **Methods:** The present study protocol settles the basis for a 16-week randomized clinical trial based on exercise prescription at the maximal fat oxidation rate combined with resistance training in young individuals with overweight and obesity. **Expected Results:** This study will elucidate how FatMax, with or without resistance exercises, can enhance metabolic flexibility, increase fat oxidation, and improve body composition, evaluating changes in biochemical parameters (cholesterol, glucose, triglycerides, and inflammatory markers), metabolic biomarkers (determination of fat and carbohydrate utilization rates during rest and exercise), and epigenetic indicators (focusing on microRNAs associated with adipogenesis, inflammation, and fat metabolism). ClinicalTrials.gov identification number: NCT06553482 (FatMax Training on Metabolic and Epigenetic Parameters).

## 1. Introduction

Obesity is a major public health challenge affecting millions of people in Mexico and around the world. This noncommunicable disease has been dramatically increased in recent decades, being recognized as one of the major risk factors for chronic diseases such as diabetes, cardiovascular disease, and certain types of cancer. By 2022, it was estimated that 890 million adults suffered from obesity [1], and the trend continues to increase independent of age, sex, and socioeconomic status.

The critical levels of this disease urgently demand the need to address this health crisis on multiple fronts. Traditional strategies to combat obesity have focused primarily on individual responsibility, emphasizing the need to improve dietary habits and increase physical activity [2]. However, reducing obesity to a mere willpower issue is insufficient to understand the complexity of the problem, which involves genetic/epigenetic, psychological, social, economic, environmental, and individual factors. Concerning lifestyle moderators, dietary counseling and exercise prescription have a profound impact on energy metabolism and body weight regulation in this population. Furthermore, distinct genetic polymorphisms and microRNA molecules have also been associated with obesity and the particular response to diet and exercise [3,4].

The current guidelines provided by the American College of Sports Medicine and the World Health Organization for obesity management emphasize that weekly energy expenditure stimulated by exercise should be higher than 2000 kcal/week to create a negative energy balance that induces body weight reduction. However, emerging research indicates that maximizing fat oxidation during exercise, rather than solely focusing on overall energy expenditure, may offer a more effective approach for the prevention and management of obesity [5]. In fact, exercise training performed at the intensity of maximal fat oxidation (FatMax) enhances metabolic flexibility and fat utilization during exercise, improving cardiovascular health in patients with obesity [6]. Additionally, FatMax training alleviates insulin resistance and systemic inflammation, decreases visceral adiposity, and improves lipid metabolism, making it a highly effective strategy for long-term weight management and metabolic health in this population [7]. FatMax training stands apart from other exercise modalities, such as HIIT and MICT, for its capacity to optimize fat oxidation as the primary energy source, which may result in greater fat loss. Furthermore, its moderate intensity makes it safer and more sustainable for overweight or obese populations, improving adherence and minimizing injury risk. In contrast, while HIIT is effective in enhancing cardiovascular fitness and reducing exercise duration, it may not be well tolerated by sedentary individuals or those with metabolic disorders [8].

It should be noticed that 8 to 20 weeks of FatMax training seem to not alter free fat mass or muscle mass in this population [7,9]. Therefore, combining FatMax training with resistance exercises may increase the lean mass of subjects with obesity, preventing sarcopenic obesity and cardiometabolic disorders related to low muscle mass over the long term [9,10]. On the other hand, it has been demonstrated that fat oxidation during exercise is determined by age, sex, body fat levels, genotype, and the type of exercise in individuals with obesity [11,12]. However, the influence of these moderators on the cardio-metabolic adaptations stimulated by FatMax training remains understudied. In addition, there is scarce information about the molecular mechanisms that become activated by FatMax training to induce positive adaptations in metabolic function and overall fitness. To be precise, Bogdanis et al. [13] and Tan et al. [14] reported that skeletal muscle mitochondrial respiratory capacity and the activity of lipoprotein lipase increased significantly after 10 weeks of FatMax training in overweight and obese individuals. Nevertheless, the impact of FatMax training on epigenetic biomarkers related to mitochondrial biogenesis and the regulation of fatty acid metabolism, such as microRNAs, remains unexplored. In this context, microRNAs such as miR-1, which regulates the expression of CPT1, a key enzyme in the transport of fatty acids to the mitochondria, and miR-145, miR-20a-5p, and miR-29a, which modulate the activity of CD36, a protein involved in the uptake of long-chain fatty acids, have been linked to the development of obesity due to their role in lipid metabolism regulation. However, exploring how these microRNAs can be modified by physical exercise interventions, such as those performed at the intensity of maximal fat oxidation (FatMax), is essential to better understand the underlying molecular mechanisms that could optimize both the prevention and treatment of obesity, thereby enhancing intervention strategies to promote overall well-being [11].

We hypothesize that FatMax training, with or without resistance exercises, will significantly improve body composition (reduce body fat and increase muscle mass) and metabolic flexibility, increase fat oxidation, and alter epigenetic markers related to adipogenesis and inflammation in overweight and obese young adults. Looking to examine which biological and lifestyle factors moderate the adaptations to FatMax training, we will investigate the influence of age, sex, body fat levels, genotype, diet, psychological variables, and the type of exercise.

## 2. Experimental Design

### 2.1. Type of Study

This research is a randomized controlled clinical trial designed to evaluate the impact of exercise training at maximal fat oxidation intensity (FatMax) combined with resistance training on metabolic, biochemical, and epigenetic parameters in young adults with overweight and obesity. The trial is set to last 16 weeks, and its design adheres to the SPIRIT standards and CONSORT guidelines to ensure rigor and replicability in the reporting process.

### 2.2. Participants Eligibility Criteria

The study will recruit 156 sedentary young adults aged between 18 and 35 years, all of whom will have a body mass index (BMI) ≥ 25 kg/m^2^. Participants will be recruited through snowball sampling. The participants will be eligible to participate in this study if they meet the eligibility criteria detailed in Table 1.

### 2.3. Interventions or Treatments

Participants will be randomly assigned to one of three groups: Group A: Will perform low-intensity training within the FatMax zone. Group B: Will perform FatMax training combined with resistance exercises. Group C (Placebo Group): The placebo group will undergo a ‘masking training’ that does not involve significant physiological or metabolic changes. This training will consist of stretching sessions with smooth, controlled movements that do not significantly impact metabolism or induce relevant physiological changes, while maintaining the perception of physical activity and an isoenergetic diet.

### 2.4. Dependent Variables

The dependent variables include body composition (fat mass and muscle mass), biochemical parameters (glucose, cholesterol), metabolic markers (fat and carbohydrate oxidation rates), and epigenetic markers (microRNAs associated with adipogenesis and fat metabolism).

### 2.5. Groups and Randomization

Randomization of participants will be conducted using SPSS 22 statistical software to ensure balanced distribution between gender and BMI categories (overweight and class 1 obesity). Participants will be randomly assigned to the three study groups, and the allocation will be coded to maintain group balance.

### 2.6. Duration and Timeline

This study will last for a total of 16 weeks, with assessments conducted at the beginning of this study (baseline) and after the 16-week intervention period to measure the changes in body composition, metabolic parameters, and epigenetic markers.

### 2.7. Control Procedures

To control potential confounding variables, participants in the control group will follow an isoenergetic diet, and dietary intake will be monitored regularly throughout this study. Individuals with less than 75% attendance at the training sessions will be excluded from the final analysis. To address potential non-compliance or dropout in this exercise intervention, a plan will be implemented that includes weekly check-ins, flexible scheduling, and the provision of motivational incentives. Close monitoring will be conducted to detect early non-compliance, with personalized support provided as needed. In cases of dropout, exit interviews will be conducted to identify barriers and improve future retention strategies. This approach aims to maximize adherence and minimize dropout throughout this study. Other variables, such as sleep quality and psychological status, will be monitored to ensure consistency across all groups.

## 3. Materials and Equipment

Materials and equipment to be used in this study is briefly described below. Their specific protocols are presented in detail in Section 4.

### 3.1. Biochemical Variables

Blood samples will be collected in EDTA vacutainer tubes and centrifuged at 3000× *g* for 20 min at refrigerated temperatures. Samples will be stored at −80 °C until analysis. Total cholesterol, c-HDL, c-LDL, triglycerides, glycosylated hemoglobin, and glucose levels will be measured using standard colorimetric procedures, following the manufacturer’s instructions (Spinreact, Girona, Spain), and analyzed with a Mindray BS200 biochemical autoanalyzer (MINDRAY, Shenzhen, China). Plasma concentrations of IL-6 and TNF-α will be assessed using Quantikine immunoassay kits (R&D Systems, Minneapolis, MN, USA).

### 3.2. Body Composition

Body composition will be assessed using a combination of techniques. Bone mineral density and total skeletal muscle mass will be measured with dual-energy X-ray absorptiometry (DXA) (GE Healthcare, Madison, WI, USA). Total body water, intracellular, and extracellular water, as well as phase angle, will be evaluated using bioelectrical impedance analysis (BIA) SECA mBCA 525 (SECA, Hamburg, Germany). Additionally, body fat mass will be measured using air displacement plethysmography (BodPod, COSMED, Rome, Italy).

### 3.3. Epigenetic Variables

For the epigenetic analysis, circulating microRNAs will be isolated from venous blood samples using the mirVana™ miRNA Isolation Kit (Thermo Fisher Scientific, Waltham, MA, USA). Fourteen microRNAs related to inflammation, adipogenesis, and fatty acid metabolism will be analyzed. Specific amplification of each microRNA will be performed using the QuantStudio 3 system (Thermo Fisher Scientific, Waltham, MA, USA), and expression levels will be quantified using the ∆∆Ct method with TaqMan MicroRNA Assays (Thermo Fisher Scientific, Waltham, MA, USA). Additionally, genotyping of polymorphisms related to obesity and lipid metabolism will be performed using TaqMan probes with a StepOnePlus thermocycler (Thermo Fisher Scientific, Waltham, MA, USA).

### 3.4. Metabolic Flexibility

To measure metabolic variables, participants will undergo submaximal exercise tests on a cycle ergometer (Monark Exercise AB, Vansbro, Sweden) or a treadmill (Quinton TM55, WA, USA). Gas exchange during exercise will be measured using a TrueOne 2400 metabolic measurement system (ParvoMedics, Sandy, UT, USA), which will be calibrated before each test. This equipment will allow for the calculation of oxygen consumption (VO_2_), carbon dioxide production (VCO_2_), and the rates of fat and carbohydrate oxidation during exercise.

## 4. Detailed Procedure

### 4.1. Sample Size Calculation

To carry out the present study, 156 young sedentary adults with overweight/obesity will be recruited through a snowball sampling. Considering the body composition changes as the primary outcome, the minimum sample size calculated for the present protocol was 44 participants to reliably detect an effect size of δ ≥ 0.5, assuming a two-sided criterion for detection that allows a maximum Type I error rate of α = 0.05 with a probability of 0.9, and considering a 20% dropout of participants, the sample size has been fixed to 52 participants per group. The statistical software used to determine the sample size was G*Power (v3.1.9.7, Universität Kiel, Düsseldorf, Germany) [15]. 

### 4.2. Randomization and Blinding

The protocol involves the assignment of 52 subjects to one of three groups. Group A will comprise participants performing low-intensity training sessions within their FatMax zone. Group B will comprise participants who will perform both FatMax zone exercises as well as muscular resistance exercises (MRE). Both groups will be compared to group C, where participants will maintain their physical activity following an isoenergetic diet, evaluated through 24 h dietary records every four weeks.

For groups A and B, subjects will be allowed to perform their FatMax training on a treadmill or stationary bicycle, whichever is of their preference. The random sequence for participant assignment will be coded with SPSS 22 Software, looking for an equal distribution of both genders, BMI (overweight (≥25 and <30 kg/m^2^) and obesity grade 1 (≥30 and <35 kg/m^2^). All the evaluations and training sessions will be conducted in the Conahcyt National Laboratory of Body Composition and Energetic Metabolism (LaNCoCoME). The general process is shown in Figure 1.

To ensure single blinding, a researcher external to this study will enter data into the software on separate datasheets so that these researchers can analyze them without having access to treatment assignment information, storing all participant files in numerical order in a secure and accessible location. All files will be retained for a period of five years.

### 4.3. Pre Intervention Exercise Test

Energy expenditure and the utilization of fat and carbohydrates at rest and during exercise will be determined by indirect calorimetry using a Metabolic Measurement System (TrueOne 2400, Parvomedics, Sandy, UT, USA), calibrated before each measurement. For basal metabolism, the subjects will remain in the Fowler anatomical position (semi-sitting supine) for continuous measurement of oxygen consumption (VO_2_) and carbon dioxide production (VCO_2_) for a period of 15 to 20 min. Subsequently, the average values of the VO_2_ and VCO_2_ recorded during a stable phase of 5 min (coefficient of variation in RQ ≤ 5%) and used to calculate energy expenditure and the macronutrient oxidation rate, using the stoichiometric equations of Weir [16] and Frayn [17], respectively.

A graded exercise protocol previously validated by the working group associated with the LaNCoCoME (Table 2) will be used to determine cardiorespiratory fitness, metabolic flexibility, and fat oxidation biomarkers. Briefly, this protocol allows an estimation of macronutrient oxidation at FatMax with a measurement error of ~1.6 g/h for carbohydrate oxidation and ~1.3 g/h for fat oxidation [18]. Distinct parameters of cardiorespiratory fitness and metabolic flexibility will also be determined by a maximal stress test exercise test on one treadmill (Quinton TM55, WA, USA) or cycle ergometer (Monark Exercise AB, Vansbro, Sweden), as described in Table 2.

Gas exchange and heart rate (Polar H10, Polar^®^ Electro OY, Kempele, Finland) will be recorded throughout the exercise tests, while the rate of perceived exertion will be registered at the end of each stage using the modified Borg scale with a 0–10 rating. The average VO_2_ and VCO_2_ values of the last 2 min of each stage before RER ≥ 1.0 will be used to calculate fat and carbohydrate oxidation using the stoichiometric equations of Jeukendrup and Wallis [19]. The maximum rate of fat oxidation and its corresponding intensity during exercise will be determined by mathematical modeling of fat oxidation kinetics, plotted against oxygen consumption (% VO_2_max) [20]. The symmetry, amplitude, and translation parameters of the curve will be calculated as well to represent the oxidative capacity of each subject.

Additionally, fat oxidation kinetics will be used to determine two new metabolic flexibility indices described by the following analytical procedures: (i) calculation of the area under the curve corresponding to fat oxidation kinetics; and (ii) simple linear regression to determine the increase in fat oxidation concerning the change in energy expenditure during the transition from rest to FatMax [21]. Both metabolic biomarkers will be calculated with GrapPad Prism v8.1 software and will be interpreted as follows: The higher the value, the better the metabolic flexibility.

The lactate and ventilatory thresholds will also be computed in ExPhysLab (https://www.exphyslab.com/, accessed on 26 October 2024), using a computerized algorithm that allows the automatic detection of these biomarkers through 25 distinct approaches [22]. Baseline blood lactate concentrations will be considered for the fitting of blood lactate kinetics by using a 3rd degree polynomial regression in all participants. Furthermore, the nine-panel approach proposed by Wasserman and Mcllroy [23] will be used to determine the ventilatory thresholds. The efficiency of oxygen consumption and the optimal breathing point will be determined by analyzing the relationship between dynamic pulmonary ventilation and oxygen consumption. The maximum oxygen consumption (VO_2_max) will be verified using the following criteria: (i) a VO_2_ plateau (increase in VO_2_ < 150 mL/min) during the final stage of the exercise protocol; (ii) RER ≥ 1.15; (iii) LA−max ≥ 8.0 mmol/L; and (iv) heart rate ≥ 90% of the theoretical maximum heart rate. The cut-off points provided by the American College of Sports Medicine will be used to determine the cardiorespiratory fitness level of the participants according to their age and sex [5]. Finally, blood lactate concentration will be measured using a portable lactate analyzer (Nova Biomedical Corp., Waltham, MA, USA). The lactate threshold will be determined by plotting the data and identifying the precise point corresponding to a 4 mmol/L concentration.

### 4.4. FatMax and Resistance Training: Frequency, Intensity, Time and Type

The characteristics of FatMax and resistance training are described in Table 3. In brief, the training sessions at FatMax will begin with a 5 min warm-up at 4 km/h (1°) or 50 w (60 rpm). Afterward, each patient will exercise on the treadmill or stationary bike for 40 min at their corresponding FatMax heart rate (±5 beats/min). The training volume will gradually rise up to 240 min/week, assessing gas exchange every four weeks to examine the acute metabolic response to prolonged exercise at FatMax and calculate fat utilization as recommended by Chávez-Guevara et al. [18].

In the case of resistance training, each session will have a warm-up phase of 5–10 min and a main physical conditioning phase of 30–40 min that will consist of developing a personalized circuit of strength exercises of the main muscle groups (arms, shoulders, chest, back, and legs) at moderate intensity, finishing with a cool-down of 5–10 min.

To reduce the possibility of molecular interference between FATmax training and resistance exercise, these sessions will be scheduled at least 6 to 8 h apart, ensuring adequate time for distinct physiological adaptations to each exercise modality.

### 4.5. Baseline and Follow-Up Measurements

#### 4.5.1. Biochemical Variables

Two blood samples will be collected per participant (8 mL after 8–10 h of fasting) during the clinical trial, the first on day 0 and the second at the end of the intervention (day 112). To ensure results that more accurately reflect the organism’s basal state, post-exercise biochemical measurements will be taken at least two days after the intervention, avoiding the temporary alterations caused by acute physical exertion. Plasma samples will be obtained by refrigerated centrifugation at 3000× *g* for 20 min and frozen at −80 °C until further analysis. Plasma samples will be processed for total cholesterol, high-density lipoprotein cholesterol (c-HDL), low-density lipoprotein cholesterol (c-LDL), triglycerides, glycosylated hemoglobin, and glucose by standard colorimetric procedures following the manufacturer’s instructions (Spinreact, Girona, Spain) with a Mindray BS200 biochemical autoanalyzer (MINDRAY, Shenzhen, China). Additionally, plasma IL-6 and TNF-α concentrations will be evaluated with the Quantikine immunoassay kit from R&D Systems (Minneapolis, MN, USA) and a sensitivity ELISA kit with an alkaline phosphatase signal amplification system, respectively (Quantikine HS, high sensitivity, R&D Systems, Minneapolis, MN, USA).

#### 4.5.2. Metabolic Variables

In addition to the metabolic flexibility outcomes described in Section 2.4, the participant’s body composition will be monitored. Briefly, the body composition of the individuals will be analyzed as follows: (i) a dual-energy X-ray absorptiometry (DXA) device (GE Healthcare, Madison, WI, USA) will be used to assess bone mineral density and total musculoskeletal mass by segments; (ii) a BIA device SECA mBCA 525 (SECA, Hamburg, Germany) will be used to evaluate total body water, intracellular, and extracellular water, as well as phase angle; and (iii) an air displacement plethysmograph (BodPod, COSMED, Rome, Italy) will be used to assess body fat mass for each participant. ISAK Level 2 and Level 3 certified anthropometrists will conduct the body measurements using certified equipment (Smartmet, Guadalajara, Mexico).

#### 4.5.3. Epigenetic Variables

To measure circulating microRNAs, peripheral venous blood samples will be collected in EDTA vacutainer tubes (5 mL). Subsequently, the mirVana™ miRNA Isolation Kit (Thermo Fisher Scientific, Waltham, MA, USA) will be used to analyze 14 microRNAs involved in inflammatory processes (miR-15, miR-29c, miR-30^a^, miR-142/3, miR-181^a^), adipogenesis (miR-188-5p, miR-146b-5p, miR-103, miR-107, miR-125b), and fatty acid metabolism (miR-33, miR-335, miR-370, miR-758).

Additionally, cDNA synthesis will be carried out through the reverse transcription of the microRNAs using the TaqMan MicroRNA-RT Kit (Thermo Fisher Scientific, Waltham, MA, USA). Specifically, amplification of the microRNAs of interest will be performed using the TaqMan Universal Master Mix II with UNG reagent (Thermo Fisher Scientific, Waltham, MA, USA). The specific amplification of each microRNA will be conducted with the QuantStudio 3 system (Thermo Fisher Scientific, Waltham, MA, USA), using probes specific to each microRNA and their internal controls through the TaqMan MicroRNA Assays MTO LG chemistry (Thermo Fisher Scientific, Waltham, MA, USA). The relative quantification of expression levels will be performed using the ∆∆Ct quantification method, validated through the quantification of amplification efficiency and the construction of a dynamic range curve.

In turn, the genetic influence on anthropometric and biochemical changes after the treatments will be evaluated using TaqMan probes (Thermo Fisher Scientific, Waltham, MA, USA) through an RT-PCR system using a StepOnePlus thermocycler (Thermo Fisher Scientific, Waltham, MA, USA). The polymorphisms to be assessed will include those related to obesity (rs9939609, *FTO*), lipid metabolism (rs1799883, *FABP2*), and physical adaptation to exercise (R577X, *ACTN3*). Genotyping will be verified using positive controls corresponding to the three possible genotypes in each experiment.

Although circulating microRNAs provide valuable insights into exercise-induced epigenetic changes, it is essential to consider the challenges related to their degradation and variability in expression. To address these issues, we will implement a comprehensive strategy to ensure microRNA stability and reduce variability in the results. Blood samples will be processed promptly, within 2 to 4 h, to separate plasma or serum, minimizing the risk of degradation. Additionally, samples will be stored at ultra-low temperatures (e.g., −80 °C) to preserve the integrity of the microRNAs over time.

#### 4.5.4. Dietary and Psychological Counseling

For internal control, covariates that could influence the findings (uncontrollable variables) will be monitored, including dietary patterns, mental health, and adherence of the individuals during the trial period. The food intake of all participants will be monitored through a 24 h quantitative recall of typical days (typical days before the start and at the end of the intervention), as well as a food frequency questionnaire. Food scales will be used to increase precision in food portions. Diet records will be analyzed for total calories, protein, carbohydrate, and fat intake (Diet Analysis Plus, ESHA Research, Salem, OR, USA). Dietary control will be complemented with biomarkers of nutrient intake such as urinary nitrogen and blood concentrations of cholesterol, triglycerides, and glucose.

The psychological status of the individuals will be evaluated using the validated instrument “Depression, Anxiety, and Stress Scale-21” (DASS-21), complemented by sleep-wake patterns obtained through a sleep diary and sleep quality assessment questionnaires (Pittsburgh Sleep Quality Index and Insomnia Severity Index).

Only individuals who complete at least 80% of all training sessions and measurements outlined in this intervention will be included in the data to ensure adherence to the treatments and the validity of the findings. Additionally, electronic and personal contact via text messaging will always be enabled for study follow-up.

#### 4.5.5. Data Analysis

The normality and homoscedasticity of the variables will be assessed using the Kolmogorov–Smirnov and Levene tests for each group. A student’s *t*-test will be used to analyze the differences between the pre- and post-treatment values for each variable if the data follows a normal distribution. If the data are non-parametric, the Mann–Whitney U test will be used. Statistical analysis between treatments will be conducted using ANOVA with Tukey’s post-hoc test in the case of parametric distribution and/or Kruskal–Wallis H and Dunn’s Post-hoc test for non-parametric distribution, with a Bonferroni setting for comparing intervention-induced changes. Additionally, Pearson and/or Spearman correlation tests will be used to establish associations between variables. Multivariate linear regression models will also be constructed to explore the contribution of treatment on main outcomes, which will be adjusted by potential confounder factors including age, sex, dietary variables, psychological variables, and baseline measurements. Moreover, multiple imputations will be used for generating replacement values for missing data. Statistical significance will be considered at *p* < 0.05, and all statistical analyses will be conducted using IBM SPSS package version 22 (SPSS Inc., IBM Company, Chicago, IL, USA) and GraphPad Prism v8.1 (GraphPad Software, La Jolla, CA, USA).

#### 4.5.6. Ethics Concerns

The clinical trial has been registered prospectively at the ClinicalTrials.gov database (Date of approval: 13 August 2024; Identification number: NCT06553482). The protocol and all its procedures have been approved by the bioethics committee of the Medical and Psychology School at the Autonomous University of Baja California (UABC) with Code D346. At all times, interested participants must sign an informed consent form, detailing this study’s objectives, benefits, risks, and discomforts related to the protocol. Anonymity and confidentiality of the data will be ensured by assigning specific codes to individuals, and no documents will link these codes with the participant’s name or other identifying information. Additionally, this study will always obey the guidelines stipulated in the Declaration of Helsinki [24].

## 5. Expected Results

International organizations widely promote physical activity and exercise to combat non-communicable chronic diseases, which are highly associated with obesity [1]. Unfortunately, most people do not meet the minimum requirements for participation in physical activity, particularly those who are overweight or obese and who express multiple limitations in adhering to recommendations. A constant barrier is the lack of knowledge on performing specific exercises [25]. In this regard, the Physical Activity Working Group of the European Association for this study of Obesity emphasized in 2021 the need for truly functional practical exercise recommendations that focus on managing overweight and obesity. It has been agreed that a moderate-intensity aerobic exercise training program promotes substantial improvements in both weight loss and the improvement of circulating lipids, total fat accumulation, and intra-visceral fat [26].

In this context, this study will elucidate how FatMax training, both alone and in combination with resistance exercises, can enhance metabolic flexibility, increase fat oxidation, and improve body composition. This study seeks to provide critical insights into the optimization of fat utilization and the reduction of visceral fat while simultaneously preserving lean muscle mass. Additionally, expected improvements in biochemical markers—such as enhanced insulin sensitivity, improved lipid profiles, and reduced systemic inflammation—will support the formulation of targeted exercise strategies to lower the risk of chronic diseases, including cardiovascular disease and type 2 diabetes. Furthermore, anticipated epigenetic changes, particularly in microRNAs associated with adipogenesis and inflammation, will offer novel perspectives on the molecular mechanisms activated by FatMax exercise. Together, these results will lay the groundwork for evidence-based, personalized exercise recommendations that address the specific metabolic and physiological needs of this population, ultimately contributing to more effective and sustainable approaches to obesity management.

If these issues are not properly addressed, the consequences will have serious public health concerns. The prevalence of obesity will continue to rise, increasing the number of people at risk of developing chronic diseases, reducing their life expectancy, and quality of life. In addition, the health care system will face increased costs and pressure, making this silent pandemic even more difficult to manage.

As limitations of this study protocol, we cannot include a double or triple blind randomized trial, due the interventions are related to a FatMax and/or resistance training program and only the data analysis researcher will be blinded. Additionally, it is acknowledged that in addition to exercise, other lifestyle factors might influence microRNA expression, such as diet, sleep quality, emotional stress, or psychological variables, so we will include them in the correlation tests and multivariate linear regression model analysis.

## Figures and Tables

**Figure 1 jfmk-09-00214-f001:**
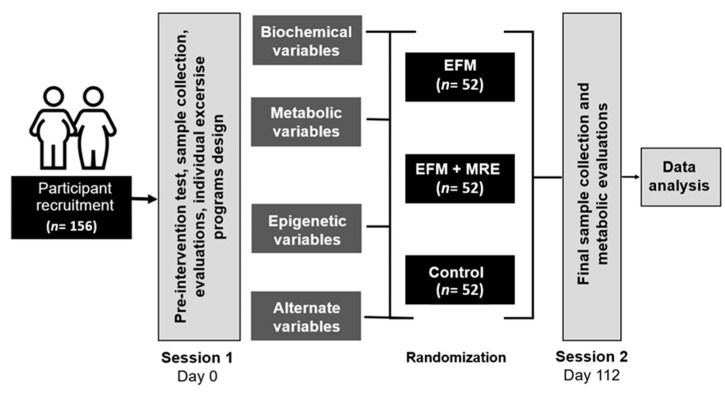
Randomized clinical trial design. EFM: FatMax training; EFM + MRE = FatMax plus low resistance muscle training. Note: All participants will follow an isoenergetic diet, and those individuals who attend less than 75% of the training sessions will be excluded from the analysis.

**Table 1 jfmk-09-00214-t001:** Eligibility criteria to participate in the clinical trial.

Inclusion Criteria	Exclusion/Elimination Criteria
Both genders	≥100 mL of alcohol consumption per week
18–35 years old	Consume a dietary supplement or pharmacological treatment
Energy expenditure < 600 METs a week	Have limitations to perform regular physical activity
Body mass index ≥ 25 kg/m^2^	Have a chronic non-communicable disease
Acceptance of informed consent	Answer “yes” to any of the questions on the Physical Activity Readiness-Questionnaire for Everyone (PAR-Q+), which represents a health risk.
	<80% attendance at exercise sessions

METs: Metabolic Equivalents of the task.

**Table 2 jfmk-09-00214-t002:** Exercise protocols for the assessment of metabolic function and cardiorespiratory conditioning.

	Treadmill	Cycle Ergometer
Warm up	Duration: 5 min Intensity: 3 km/h; 0% inclination	Duration: 5 min; 40 Watts; 60 RPM
Phase 1	Duration of stages: 3 min Initial load: 3 km/h; 1% inclination Progression of the load: 1 km/h Conclusion: RER = 1.0 for 30 s	Duration of stages: 3 min Initial load: 40 Watts; 60 RPM Progression of load: 20 Watts Conclusion: RER = 1.0 for 30 s
Phase 2	Duration of stages: 1 min Load progression: 1 km/h Conclusion: (i) Rate of perceived exertion = 10; (ii) Heart rate ≥ 90% MHR	Duration of stages: 1 min Load progression: 20 Watts Conclusion: (i) Rate of perceived exertion = 10 Effort = 10; (ii) Heart rate ≥ 90% MHR

MHR: theoretical maximum heart rate; RER: respiratory exchange rate; RPM: revolutions per minute.

**Table 3 jfmk-09-00214-t003:** FatMax and Resistance Training characteristics.

Training	Frequency	Intensity	Time	Type
FatMax	Fout times a week	Individualized according to the Heart Rate at thetMax	Sessions between 1–1.5 h	Aerobic exercise
Resistance Training	Two times a week	Individualized, with a total of 3 sets of 8–12 reps per muscle group a week	Sessions between 40–60 min	Resistance training (weight lifting)

## Data Availability

Any researcher that contacts the first author, M.A.H.-L. (marco.antonio.hernandez.lepe@uabc.edu.mx), will have access to this study data required.

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
