# Peer review of "Impact of Exercise Training at Maximal Fat Oxidation Intensity on Metabolic and Epigenetic Parameters in Patients with Overweight and Obesity: Study Protocol of a Randomized Controlled Trial"

_jfmk, 2024, doi:10.3390/jfmk9040214_

Round 1
Reviewer 1 Report (Previous Reviewer 2)
Comments and Suggestions for Authors
I would like to thank for the opportunity to review this manuscript. Please see the following comments to consider to further improve the quality of this manuscript.
While the introduction covers the background, the hypothesis isn't explicitly stated. Clearly define the primary and secondary hypotheses, making the expected effects of FatMax and resistance training on metabolic, biochemical, and epigenetic markers more explicit. Add a section at the end of the introduction clearly stating the study’s hypotheses. For example: "We hypothesize that FatMax training, with or without resistance exercises, will significantly improve metabolic flexibility, reduce body fat, and alter epigenetic markers related to adipogenesis and inflammation."
The manuscript mentions the use of G*Power software for sample size calculation, but it lacks details about the effect sizes and the assumptions behind the calculation. Include details about the assumptions made for effect size, power, and expected variance in outcomes to justify the sample size more comprehensively. This will enhance the transparency of the methodology.
While randomization is addressed, blinding procedures are not sufficiently covered, which is essential for reducing bias. Provide more details about whether the researchers performing the outcome assessments will be blinded to group allocation. If blinding is not possible, explain how potential biases will be minimized.
Although the statistical methods are briefly mentioned, the description could be more detailed, particularly regarding handling potential confounders or missing data. Elaborate on how potential confounders (e.g., baseline differences in BMI, age, and sex) will be controlled in the analysis. Specify how missing data will be handled (e.g., intention-to-treat analysis, multiple imputation).
The control of diet is mentioned but seems to rely on self-reported 24-hour recalls, which can be prone to bias. Consider supplementing dietary control with more objective measures such as periodic checks of biomarkers (e.g., urinary nitrogen for protein intake) or providing standardized meals during key periods of the study to ensure consistency.
The manuscript discusses the role of miRNAs but does not fully explore the potential variability in miRNA expression due to factors other than exercise, such as sleep and stress. Acknowledge these limitations in the discussion section and consider incorporating additional controls (e.g., questionnaires or biomarkers for stress and sleep quality) to account for these factors.
Author Response
October 24th, 2024
We truly appreciate your contributions. The changes requested can be identified in the new version of our manuscript (jfmk-3295557-R1) while a response to your comment is described below:
I would like to thank for the opportunity to review this manuscript. Please see the following comments to consider to further improve the quality of this manuscript.
While the introduction covers the background, the hypothesis isn't explicitly stated. Clearly define the primary and secondary hypotheses, making the expected effects of FatMax and resistance training on metabolic, biochemical, and epigenetic markers more explicit. Add a section at the end of the introduction clearly stating the study’s hypotheses. For example: "We hypothesize that FatMax training, with or without resistance exercises, will significantly improve metabolic flexibility, reduce body fat, and alter epigenetic markers related to adipogenesis and inflammation.”
R= We appreciate your observation, to define clearly the hypotheses of our study protocol, it has been added the following sentence in the final paragraph of the Introduction Section: “We hypothesize that FatMax training, with or without resistance exercises, will significantly improve body composition (reduce body fat and increase muscle mass) and metabolic flexibility, increase fat oxidation, and alter epigenetic markers related to adipo-genesis and inflammation in overweight and obese young adults. Looking to examine which biological and lifestyle factors moderate the adaptations to FatMax training, we will investigate the influence of age, sex, body fat levels, genotype, diet, psychological variables, and the type of exercise”.
The manuscript mentions the use of G*Power software for sample size calculation, but it lacks details about the effect sizes and the assumptions behind the calculation. Include details about the assumptions made for effect size, power, and expected variance in outcomes to justify the sample size more comprehensively. This will enhance the transparency of the methodology.
R= According to your observation, we added the following statement about Sample size calculation in Section 4.1: “Considering the body composition changes as primary outcome, the minimum sample size calculated for the present protocol was 44 participants to reliably detect an effect size of δ≥0.5, assuming a two-sided criterion for detection that allows a maximum Type I error rate of α=0.05 with a probability of 0.9, and considering a 20% dropout of participants, the sample size has been fixed to 52 participants per group”. Thank you.
While randomization is addressed, blinding procedures are not sufficiently covered, which is essential for reducing bias. Provide more details about whether the researchers performing the outcome assessments will be blinded to group allocation. If blinding is not possible, explain how potential biases will be minimized.
R= We have added the following sentence related to the blinding procedure in Section 4.2: ” To ensure single blinding, a researcher external to the study will enter data into the software on separate datasheets so that the researchers can analyze them without having access to treatment assignment information, storing all participant files in numerical order in a secure and accessible location. All files will be retained for a period of five years”. It is important that we have added an explanation of the limitations of the blinding procedures in the last paragraph of the manuscript (Lines 442-444), including the following sentence: “As limitations of this study protocol, we can’t include a double or triple blind randomized trial, due the interventions are related to a FatMax and/or resistance training program and only the data analysis researcher will be blinded”.
Although the statistical methods are briefly mentioned, the description could be more detailed, particularly regarding handling potential confounders or missing data. Elaborate on how potential confounders (e.g., baseline differences in BMI, age, and sex) will be controlled in the analysis. Specify how missing data will be handled (e.g., intention-to-treat analysis, multiple imputation).
R= We have detailed better the statistical methods related to potential confounders in Section 4.5.5, including the following sentence: “Multivariate linear regression models will also be constructed to explore the contribution of treatment on main outcomes, which will be adjusted by potential confounder factors including age, sex, dietary variables, psychological variables, and baseline measurements. Moreover, multiple imputation will be used for generating replacement values for missing data”. We appreciate your observation.
The control of diet is mentioned but seems to rely on self-reported 24-hour recalls, which can be prone to bias. Consider supplementing dietary control with more objective measures such as periodic checks of biomarkers (e.g., urinary nitrogen for protein intake) or providing standardized meals during key periods of the study to ensure consistency.
R= According to your observation, we have added the evaluation of variables and biomarkers accessible to our research group, resulting in the following sentence: “Food scales will be used to increase precision in food portions. Diet records will be analyzed for total calories, protein, carbohydrate, and fat intake (Diet Analysis Plus, ESHA Research, Salem, OR). Dietary control will be complemented with biomarkers of nutrient intake such as urinary nitrogen, and blood concentrations of cholesterol, triglycerides, and glucose”.
The manuscript discusses the role of miRNAs but does not fully explore the potential variability in miRNA expression due to factors other than exercise, such as sleep and stress. Acknowledge these limitations in the discussion section and consider incorporating additional controls (e.g., questionnaires or biomarkers for stress and sleep quality) to account for these factors.
R= We have added a paragraph at the final of the manuscript with the limitations of the study, where we included the following sentence related to your observation in Lines 444-447: “Additionally, it is acknowledged that in addition to exercise, other lifestyle factors might influence microRNA expression such as diet, sleep quality, emotional stress or psychological variables, so we will include them in the correlation tests and multivariate linear regression models analysis”.
According to your observations, the article has been improved substantially. We really appreciate your contributions.
Dr. Francisco Javier Olivas-Aguirre, Corresponding author
Reviewer 2 Report (Previous Reviewer 1)
Comments and Suggestions for Authors
Thanks for addressing my comments on the first submission.
Author Response
Thank you very much.
Round 2
Reviewer 1 Report (Previous Reviewer 2)
Comments and Suggestions for Authors
Authors have done well job on revising their manuscript.
This manuscript is a resubmission of an earlier submission. The following is a list of the peer review reports and author responses from that submission.
Round 1
Reviewer 1 Report
Comments and Suggestions for Authors
The manuscript has focus on the methodology of a physical intervention study in young cohort with BMI over 25 kg/m2. The proposed clinical trial has already ethical approval and registration at ClinicalTrials.gov
I will only provide some suggestions that the researchers may want to consider.
The methodology is not clear on the allowed post-exercise intake of foods and drinks which could affect parameters of interest.
The combination of resistance training and the exercise at FATmax is concurrent training in which molecular interference is possible when both exercise modalities are performed within a few hours in one session. Molecular interference can affect adaptations of the parameters of interest. The protocol is not clear whether concurrent training will happen in one session.
L273. It is advised to take blood samples at least 2 days on completion of the last exercise training session.
L296. The authors need to consider the exact time points for collection of blood samples for epigenetic variables to avoid observations that are due to the last exercise training session.
Reviewer 2 Report
Comments and Suggestions for Authors
I would like to thank for the opportunity to review this manuscript. Please see the following comments to further improve the quality of this manuscript.
While the introduction covers the significance of obesity, the rationale for selecting FatMax training could be strengthened by discussing why it may be superior to other exercise modalities, beyond mentioning energy expenditure. Add a more detailed explanation of why FatMax training specifically was chosen, particularly in comparison to other exercise regimens like HIIT or moderate-intensity continuous training (MICT). This will help readers understand the uniqueness of the study.
The discussion on epigenetic markers (microRNAs) related to obesity is concise but could benefit from a more in-depth review of the latest findings on exercise-induced epigenetic modifications. Expand the literature review to include more recent studies on how exercise interventions like FatMax impact epigenetic markers, particularly microRNAs. This would strengthen the background and rationale for including these biomarkers in the study.
The control group follows an isoenergetic diet but does not engage in any form of exercise. Consider adding a “placebo” activity for the control group, such as light stretching, to account for any psychological effects of participation. This would help isolate the physiological effects of the FatMax and resistance training from those of simply being part of a study.
The manuscript mentions that participants with less than 75% attendance will be excluded, but it’s unclear how adherence to dietary recommendations and exercise intensity will be objectively monitored. Clarify how adherence to both the exercise regimen and dietary intake will be monitored (e.g., through wearable technology, food logs, or regular check-ins). Additionally, include a plan for addressing non-compliance or dropout, as this is common in exercise trials.
While the analysis of microRNAs is clearly described, the potential challenges of isolating these markers from blood samples and ensuring their stability are not discussed. Include a brief note on the limitations of measuring circulating microRNAs (e.g., degradation, variability in expression), and how the study will control for these potential confounders.
The manuscript briefly mentions monitoring sleep and psychological status but does not elaborate on how these variables will be controlled in the final analysis. Provide more detail on how potential confounding factors like sleep quality, stress levels, and mental health will be accounted for in the statistical analysis, as these factors can influence metabolic and epigenetic outcomes.